# Heart Failure Evolution Model Based on Anomalous Diffusion Theory

**DOI:** 10.3390/e24121780

**Published:** 2022-12-06

**Authors:** Andrzej Augustyn Walczak

**Affiliations:** Faculty of Cybernetics, Military University of Technology, Gen. Kaliskiego St. 2, 00-908 Warsaw, Poland; andrzej.walczak@wat.edu.pl

**Keywords:** anomalous diffusion, disease evolution, non-Gaussian process

## Abstract

The unexpectable variations of the diagnosed disease symptoms are quite often observed during medical diagnosis. In stochastics, such behavior is called “grey swan” or “black swan” as synonyms of sudden, unpredictable change. Evolution of the disease’s symptoms is usually described by means of Markov processes, where dependency on process history is neglected. The common expectation is that such processes are Gaussian. It is demonstrated here that medical observation can be described as a Markov process and is non-Gaussian. Presented non-Gaussian processes have “fat tail” probability density distribution (pdf). “Fat tail” permits a slight change of probability density distribution and triggers an unexpectable big variation of the diagnosed parameter. Such “fat tail” solution is delivered by the anomalous diffusion model applied here to describe disease evolution and to explain the possible presence of “swans” mentioned above. The proposed model has been obtained as solution of the Fractal Fokker–Planck equation (FFPE). The paper shows a comparison of the results of the theoretical model of anomalous diffusion with experimental results of clinical studies using bioimpedance measurements in cardiology. This allows us to consider the practical usefulness of the proposed solutions.

## 1. Introduction

In the history of a disease, there is often talk of an unpredictable reaction of the patient’s body. In the language of statistics, we would expect a precise indication of whether to consider sudden changes in the evolution of a disease as the equivalent of a “black swan” or a “grey swan” [1]. In terms of disease evolution, the observed change of disease symptoms is assumed. The basic classification into “grey and black” swans is based on two elements—the predictability of their occurrence and the consequences of their occurrence. The effects of the occurrence of a “grey swan” are known and are assumed to be predictable with an unknown value of variation. The consequences of a “black swan” occurrence are classified as catastrophic and unpredictable [1]. 

What each of these cases has in common is that the variation of the quantity characterizing the variation of the observed parameter is a power function of time
(1)$$\langle x^{2}\left(t\right)\rangle =K_{\alpha }t^{-\alpha }$$
or, in Hurst exponent notation
(2)$$\langle x^{2}\left(t\right)\rangle =K_{H}t^{-2H}$$
with exponent α > 0 [2]. 

Assuming that *H* is not equal to 1/2, it is interpreted in the literature as the presence of long-time correlations as in Madelbrot work [3]. When *H* is not equal to 1/2, but there are no autocorrelations, then it is a Levy process with infinite variation [4].

More generally, this is captured by the density of the probability distribution characterized by so-called “fat tails”, which means that even a slight change in the value of the probability density function corresponds to a significant change in the variation of the observed variable [1]. Thus, the grey and black swans are governed by a stochastic process, in which a small change in the value of the probability density function is associated with an unexpectedly large variance of the parameter because the probability distribution of the observed parameter has a “fat tail,” that is, it is not described by a Gaussian process. 

Medical cases, due to their relative predictability of effects, can be classified (if they occur) as “grey swans.” However, it remains an open question whether stochastic processes that characterize the behavior of diagnosed medical parameters can have “fat tails.” If so, the occurrence of “grey swans” would be expected and frequent or even constantly present.

To study the nature of the stochastic process governing changes in medical parameters by observing their variance, one needs to have a large population of observations recorded under the same clinical conditions [5]. This is an extremely long and expensive process even with the current state of IT (meaning computer science) solutions in medicine.

We have proposed in earlier publications a different analysis using heart failure cases as an example [6,7]. Since we do not have a very large population of measurements of the selected parameter to be diagnosed (a few thousand measurements made under the same conditions of clinical observation of the patient, which guarantees their statistical utility but is a relatively small population), we checked whether, in a smaller population, the time series of observations is or is not well approximated by a Markov process. A Markov process, if it approximates well the results of a clinical observation, can indicate that the recorded observable is subject to a Gaussian stochastic process. This is a very rough estimate. Bassaler and co-authors showed that even when process variation would indicate the existence of “fat tails” in the probability density distribution, the processes can be, and most often are, Markov processes. This means that there is no need to consider the effect of history on the current value of the recorded observable [8]. The aforementioned authors proved that the determination of the value of the Hurst exponent, without further demonstration that the increments of the recorded observable are stationary or that the observed dynamics of the process directly depend on the history, is not a basis for claiming the existence of autocorrelation in the process. The admissibility of a Markov process with a Hurst exponent not equal to ½ implies the existence of non-stationary increments in the Markov process. This would mean that a good fit of the experimental results to the Markov process does not rule out non-stationary changes in the process and does not rule out that the observed process is nevertheless a process with “fat tails.”

Our aim is to find out if observed processes in disease evolution have “fat tails” or, in other words, if the Hurst exponent for registered observations is different from H = 1/2. This is because our experiment was conducted in the regime of exacting, multicenter clinical observations. 

In the following, we propose to extend the analysis of disease evolution based on the fractal F–P equation and its analytical solutions. Fitting to experimental data allows us to verify whether the non-Gaussian process can be an effective model of disease evolution. The analysis presented here consists of experimental and theoretical parts. In the experimental part of the project, 605 patients with heart failure were recruited. The study was a multi-site, randomized, controlled group trial conducted in multiple centers in parallel (ClinicalTrials.gov Identifier: NCT03476590). The study included hemodynamic parameters measured by bioimpedance techniques and electrocardiography in parallel. The same experimental results were used in earlier work, which allows a common reference to the relationship between theoretical models and clinical observations.

## 2. Materials and Methods

### 2.1. Experimental Data

The experimental data used in this study were sourced from the AMULET research project [9,10,11,12], under which a multicenter, prospective, randomized, open-label and controlled, parallel group trial was conducted (ClinicalTrials.gov Identifier: NCT03476590) at nine locations throughout Poland. In total, 605 patients with heart failure were recruited. In our study, we examined electrocardiogram (ECG—electrocardiogram) and impedance (ICG— impedance cardiogram) curves recorded with the use of an ICG device (Cardioscreen 2000, Medis, Illmenau, Germany). This non-invasive diagnostic method allows one to collect a set of specific hemodynamic parameters, such as heart rate (HR—heart rate), stroke volume (SV—stroke volume), and thoracic fluid content (TFC—thoraric fluid content). In our analysis, TFC has been used as an illustrative example. The value of TFC is the inverse of chest impedance with unit (1/Ohm) and strongly depends on fluid content of the chest. The TFC is basic information used in the diagnosis of heart failure; because of fluid overload in the pathogenesis of HF exacerbation, early detection of fluid retention is of key importance in preventing emergency admissions. For such reason, our analysis took TFC into account as the most important parameter. TFC values were measured during a clinical examination of 605 patients, performed in a relaxed, seated position. The total number of measurements recorded here is 2860, with the number of individual patient observations differing for each patient within the measurement set. The results registered are illustrated in Figure 1. At least half of the patients were investigated several times, with the observation period lasting for up to 12 months per patient. All intervals of the investigated TFC value have been divided arbitrarily into 15 discrete value slots, as shown in Figure 1. The assumed number of slots must be greater than five to allow statistical analysis (especially χ^2^ tests) but not too high to avoid complexity of calculations. With computer calculation, we estimate that the amount of 15 slots allows for an acceptable accuracy of calculations. The time series was registered as follows: Due to the irregular flow of the measured data per patient, we adopted a registration period that was 31 days long for each slot. Each slot includes hits counted during measurements registered over a five-month period. Longer registration does not carry new results in TFC distribution. Hits registered during each single measurement period are illustrated in Figure 1.

One can observe that, for the increasing ∆t, the total number of TFC hits registered tends to have the same stationary distribution of TFC, as shown in Figure 1. We assumed that the period of 5 months during which the measurements were made was sufficient for proper estimation of stationary distribution.

### 2.2. Theoretical Models 

As mentioned, two theoretical models—the Markov process [6] and the analytical solution of the Fokker–Planck equation (FPE) of diffusion containing a drift component for modeling the presence of therapy [7]—were previously studied in parallel. 

The results described in papers [6,7] indicated good agreement between experimental observations and theoretical models, confirming to some extent that the observed processes can be considered Gaussian processes. In the following, we examine anomalous diffusion to describe the distribution of observed hemodynamic parameters to show that the process may nevertheless have “fat tails.”

Anomalous diffusion is found in a wide variety of such systems, with non-linear growth of the mean squared displacement, observed over the course of time, being its hallmark. The above means that process x(t) complies with the rule, which is common in a diverse number of complex systems [13,14,15,16,17,18,19,20,21,22,23].

In order to obtain a more general and adequate class of FPE solutions, it is necessary to use a fractional FP equation (FFPE) to describe a complex system [24,25,26,27,28,29,30,31,32]. This is particularly true in the context of continuous time random walk (CTRW) and anomalous diffusion. The procedure requires the inclusion of a non-Debye relaxation in the process of changing from one slot of the measured value to another. Debye relaxation fulfils the following rule [33,34,35].
(3)$$\frac{df\left(t\right)}{dt}=-\frac{1}{\tau }\;f\left(t\right)\;,t>0$$

The following is an integral form of (3):(4)$$f\left(t\right)-f_{0}=-\frac{1}{\tau }\frac{d^{-1}f\left(t\right)}{dt^{-1}}=-\frac{1}{\tau }\int_{0}^{t}dt^{\prime }\;f\left(t^{\prime }\;\right)$$

Generalization from an integer to a rational derivative form requires the following transformation:(5)$$f\left(t\right)-f_{0}=-\frac{1}{\tau^{\alpha }}\frac{d^{-\alpha }f\left(t\right)}{dt^{-\alpha }}=\{\begin{array}{c} \frac{1}{\Gamma_{Euler}\;\left(\alpha \right)}\int_{0}^{t}dt^{\prime }\frac{f\left(t^{\prime }\right)}{\left(t-t^{\prime }\right)},\;\alpha >0\\ f\left(t\right),\;\;\;\alpha =0 \end{array}$$

In the case of non-Debye processes, the Fractal Fokker–Planck Equation (FFPE) is of the form: (6)$$\frac{\partial P\left(x,t\right)}{\partial t}=D_{t}^{1-\alpha }K_{\alpha }\frac{\partial^{2}}{\partial x^{2}}\;P\left(x,t\right)$$

In (6), the Riemann Liouville fractional differintegration is of the same form as in the non-Debye diffusion model above:(7)$$D_{t}^{1-\alpha }=\frac{\partial }{\partial t}D_{t}^{-\alpha }\frac{1}{\Gamma \left(\alpha \right)}\frac{\partial }{\partial t}\int_{0}^{t}dt^{\prime }\frac{1}{{\left(t-t^{\prime }\right)}^{1-\alpha }}$$

Finally, we should write:(8)$$\frac{\partial^{\alpha }f\left(t\right)}{\partial t^{\alpha }}=\frac{1}{\Gamma \left(\alpha -n\right)}\frac{\partial^{n}}{\partial t^{n}}\int_{0}^{t}dt^{\prime }\frac{f\left(t^{\prime }\right)}{{\left(t-t^{\prime }\right)}^{\left(1+\alpha -n\right)}}$$

With *n* −1 ≤ α < *n*, and the propagator (Green function) of (6), we obtain the following form:(9)$$P_{\alpha ,K_{\alpha }}\left(x,t,x_{0}\right)=\frac{1}{2\sqrt{\frac{\Gamma \left(\alpha +1\right)K_{\alpha }}{2}}}f_{\beta ,\gamma }\left(t,\;\frac{\left|x-x_{0}\right|}{\sqrt{\frac{\Gamma \left(\alpha +1\right)K_{\alpha }}{2}}},\right)$$
(10)$$f_{\beta ,\gamma }\left(t,\;g\right)=\frac{1}{t^{1+\beta }}\sum_{k=0}^{\infty }\frac{1}{k!\;\Gamma \left(-k\gamma -\beta \right)}{\left(-\frac{g}{t^{\gamma }}\right)}^{k}$$
(11)$$\beta =\frac{\alpha }{2}-1\;,\gamma =\frac{\alpha }{2},g=\frac{\left|x-x_{0}\right|}{\sqrt{\frac{\Gamma \left(\alpha +1\right)K_{\alpha }}{2}}}$$

The solution of FFPE (9) allows value classes of functions *P*(*x*, *t*) that exceed 1 and that may also be lower than 0. So, for *P*(*x*, *t*) to be interpreted as a probability density function (pdf), one must adopt specific constraints for *α* and *K_α_* to ensure proper pdf interpretation. The constraint applies just to solution classes in Equation (9). Now, one may simulate the obtained solution (9–11), which must obey the predefined constraint 0 ≤ *P*_*α*,*Kα*_ (*x*,*t*) ≤ 1 for a set of pairs {*α*, *K_α_*}. The general constraint of 0 < *α* < 2 applies, as a rule, for fractional FFPE. The condition that *P*(*x*, *t*) is a decreased function of both arguments also imposes a constraint on the permissible pairs {*α*, *K_α_*}. Figure 2 presents an example of the simulated *P*_*α*,*Kα*_ (*x*,*t*) with arbitrary values of *α* and *K_α_*.

We use the Green’s function of the FPE Equation (9) to calculate the time evolution of the probability density distribution for TFC starting from the selected stage of the stationary distribution obtained during the recording of the results and shown in Figure 1.
(12)$$P_{\alpha ,\;K_{\alpha }}\left(x,t\right)=\int_{0}^{\infty }P_{\alpha ,\;K_{\alpha }}\left(x,t,\;x^{\prime }\right)\;TFC\;\left(x^{\prime }\right)dx^{\prime }$$

## 3. Results

The results of the fit between the theoretical model and clinical observations are presented in Figure 3.

Values of α and K_α_ placed inside Figure 3 have been calculated by means of root mean square error (RMSE) between measured experimental data and the theoretical “fat tail” distribution presented in Equation (12), closed for a time of observation as long as 5 months. One can see that the Hurst exponent is equal to α/2 = H = 0.235. 

## 4. Discussion

The observed experimental results may depend on the ongoing therapy that regulates TFC. In other words, subjecting a patient to therapy does not rule out the existence of a low or very low dependence of the observed parameters on their history, that is, it does not rule out a possible temporal correlation. However, the effect of history on the observation performed during therapy, if any, appears to be negligibly small. This is indicated by the low value of the Hurst exponent (H = 0.235).

Importantly, it is both the indicated previous work based on Markov process models and the present model based on anomalous diffusion that effectively approximate the course of disease evolution. Thus, we do not settle on which process—Gaussian or non-Gaussian—is a better approximation. In view of the work of Besseler and co-authors as well as results presented here, it should be assumed that the process of disease evolution is a process with “fat tails”, but the use of Markov models is perfectly acceptable and effective, and the existing temporal correlation in the observed experiment is weak. This may be caused by the therapy conducted during the observation of patients, which significantly affects the temporal correlations in the evolution of the disease.

One of the peculiarities of observing disease symptoms is that we do not analyze the evolution of the disease as its effect on the patient’s body. There are always three elements: the therapy used, the patient’s body responding to the therapy, and the evolution of the therapy-treated disease in the patient’s body. The evolution of any disease described in the medical literature usually has such an entangled relationship with both the applied therapy and the patient’s body’s reaction to the disease and therapy simultaneously. When we talk about the evolution of a disease, we are describing and measuring the effects of the interaction of the aforementioned three elements, and the stochastic process we are analyzing concerns the temporal evolution of the combination of these three elements. We do suppose that such complicated relations cause unexpected jumps of registered disease symptoms. In fact, disease observation is a registration of system dynamics composed with three interacting parts. So, non-Gaussian processes seem to be expected in such systems.

The results obtained, which indicate that the stochastic process describing the evolution of a disease can have “fat tails,” show at the same time that “grey swans” in the observation of a patient’s condition are a natural phenomenon and can occur often. If the process were Gaussian, then their occurrence would be rather unexpected, and it is known from medical practice that they are a fairly common part of clinical observations. Finally, we achieve the aim of the investigation and prove that “fat tail” probability distribution is present in the disease symptoms’ observation. 

## Figures and Tables

**Figure 1 entropy-24-01780-f001:**
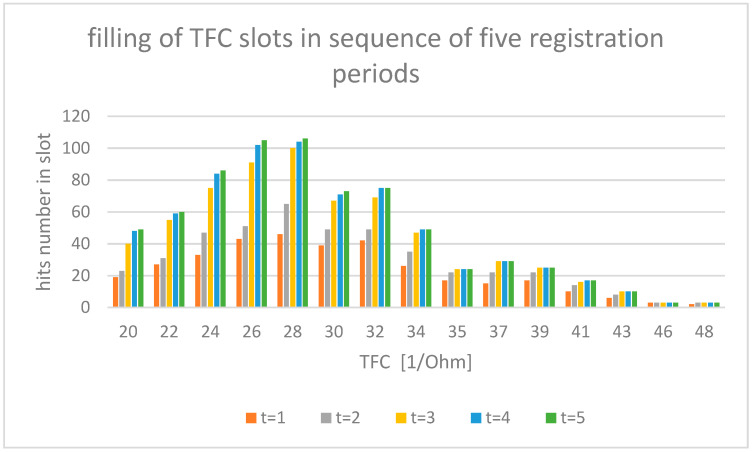
Measured number of hits of TFC values for each period of registration. Items from t1 to t5 mean one month of hits registration indicated by bar color in the figure.

**Figure 2 entropy-24-01780-f002:**
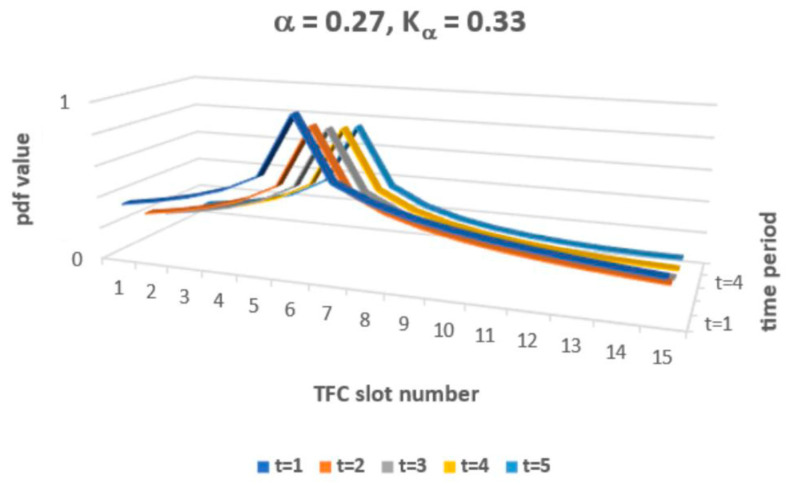
Theoretical *P*(*x*, *t*) shape for selected *α* and *K_α_*.

**Figure 3 entropy-24-01780-f003:**
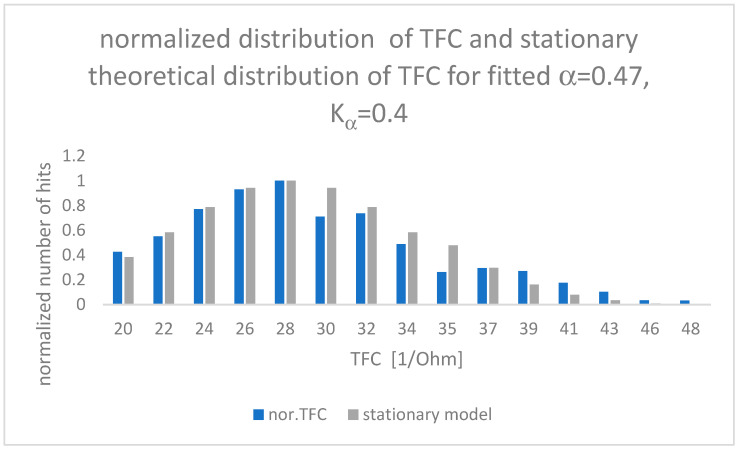
Comparison of TFC clinical observations (blue) and theoretical results (grey) for parameters α and K_α_ minimizing RMSE error value (RMSE = 0.003).

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
