# Peer review of "Heart Failure Evolution Model Based on Anomalous Diffusion Theory"

_entropy, 2022, doi:10.3390/e24121780_

Round 1

Reviewer 1 Report

The topic can be interesting but the way in which it was presented needs to be improved. The initial assumptions on which the model is based could be better explained; the description of the model implementation and of the obtained results has to be more rigorous and each step of the followed procedure should be described more in detail. This will allow to the reviewer to do a reliable evaluation of the proposed work and of its value.

The organization of the paper should be completely revised in order to improve the clarity of the text. After introduction, a section dedicated to the description of the database used and of the followed procedure should be present, then a section dedicated to the obtained results and eventually a discussion to comment the procedure and the results.

Eventually, the authors should better emphasize the advantages (including clinical utility) and also the limitations of the implemented model.

Below my specific comments:

·        In general, the followed procedure should be described more in detail and the obtained results based on quantitative evaluation.

·        In the introduction, the contextualization should be performed not only for the methodological part but also for the clinical part (that seems to be minimal in this case), being described a biomedical application.  

·        Lines 77-82 should be better argued. Is the author referring to an already performed study or to the work they are going to present?

·        At the end of introduction, I suggest clearly expressing the aim of the study.

·        At line 83 the author said the analysis they performed consisted of experimental and theoretical parts. The two parts are not clearly described and justified.

·        What does it mean that, in the analysis, TFC has been used as an illustrative example? An illustrative example of what? Thus, were HR and SV simply mentioned and then not considered? It is not clear what are the features used to implement the model.

·        Figure 1 should be better explained: for example, what does each bar stand for? What does each color means? What does “t” mean? I suggest not taking anything for granted. This will help the correct interpretation of the figure.

·        What does the author conclude from the experimental analysis showed in figure 1? For example, the author said “We assumed that the period of 5 months during which the measurements were made is sufficient for proper estimation of stationary distribution”: from what can they deduce this? Moreover, where is the indication of months in the figure? And eventually, were the observations deduced from the experimental evaluation based on qualitative considerations?

·        Not all the equations are mentioned in the text. As figures and tables, they should be mentioned and described in the text and then, inserted.

·        It seems that many aspects were taken for granted.

·        I suggest reconsidering the last section: I suggest not calling it summary if it is the discussion section and not inserting a summary section since there is the abstract.

·        The value of the Hurst exponent for the proposed model cannot be mentioned only at the end of the work.

·        Quality of figures can be improved.

·        How does the discussion at lines 207-215 influence the implementation of the proposed model?

·        What are the main different results obtained by the Markov process models and by the model based on anomalous diffusion? What is the added value given by the proposed model with respect the previously proposed ones?

·        All the acronyms have to be defined at their first use (the acronym IT, line 56, is not defined, as well as the acronym “FPE” at line 138).

·        What does it mean “pdf” at line 172?

Author Response

Answer for the referee comments

{-> each answer is placed in brace bracket in green}

The topic can be interesting but the way in which it was presented needs to be improved.

  • The initial assumptions on which the model is based could be better explained; {-> text in introduction has been revised and assumptions described more precisely}
  • the description of the model implementation and of the obtained results has to be more rigorous and {-> it seems to the author that model implementation is comprehensive. If “fat tails” distribution is a rule in medical observations then unexpected jumps of the diagnosed parameters are allowed. So proper tool for prediction of the disease evolution should be based on anomalous diffusion as it was illustrated.}
  • each step of the followed procedure should be described more in detail. {-> the assumed explanation was organized in rather ordinary way: introduction to show what is the task resolved in publication and why such task is analysed, experimental data based on strong clinical observations (proper and valuable publications are cited) made with steady participation of author, the next are details of theoretical model with explanation how one can get the same result as solution of Fractal Fokker-Planck equation, the next fit between experimental data and obtained theoretical model of anomalous diffusion to find out if such model is proper to describe clinical observation, finally conclusions from accordance of the theory and experiment. It is hard to add further details. So if some of them are not clear enough be so kind please and ask more precisely.}

This will allow to the reviewer to do a reliable evaluation of the proposed work and of its value.

  • The organization of the paper should be completely revised in order to improve the clarity of the text. After introduction, a section dedicated to the description of the database {-> what database? In the paper part of results obtained during clinical observations was exploited. All are signal registration, so it is quite obvious that NoSQL database was used in “information part of the experiment”. But it is not the aim of the paper to describe measurement system applied in clinical observations. It was presented in cited articles. So be so kind please and explain the database about which You are thinking?} used and of the followed procedure should be present {->followed procedure is presented in form of experimental data for chosen, illustrative medical parameter followed by fit between theoretical model and that data, so what do you have in mind as “followed procedure”?, } then a section dedicated to the obtained results and eventually a discussion to comment the procedure and the results {-> such discussion is in the paper}.
  • Eventually, the authors should better emphasize the advantages (including clinical utility) and also the limitations of the implemented model.{-> text in conclusions have been revised and slightly rearranged to underline clinical utility though journal is Entropy, so paper is near to entropy and stochastics than medicine}

Below my specific comments:

  • In general, the followed procedure should be described more in detail and the obtained results based on quantitative evaluation. {-> in the figure 3 are results of fit for experimental data and theoretical calculations as well as RMSE value. What other quantitative evaluation do You expect?}
  • In the introduction, the contextualization should be performed not only for the methodological part but also for the clinical part (that is minimal in this case), being described a biomedical application. {-> role of predictability has been added. Thank You for that comment
  • Lines 77-82 should be better argued. Is the author referring to an already performed study or to the work they are going to present?{-> the first sentence in line 77 is “..in the following... “. Such sentence seems to me completely evident as informing, which denotes further part of the presented paper}
  • At the end of introduction, I suggest clearly expressing the aim of the study.{->done. Thank You for this comment}
  • At line 83 the author said the analysis they performed consisted of experimental and theoretical parts. The two parts are not clearly described and justified.{-> I can understand this comment. In part 2 (lines 90-124) all experiment was presented as well as couple of citations in valuable journals with deep explanations of medical part of investigation which was exploited in the paper. In part 3 (lines 125-190) detailed explanation of theoretical approach was placed. What is not described and justified? Be so kind please and ask more precisely}
  • What does it mean that, in the analysis, TFC has been used as an illustrative example? An illustrative example of what? Thus, were HR and SV simply mentioned and then not considered? It is not clear what are the features used to implement the model.{ -> exactly as it was write – “.. illustrative examples od data..”. It is quite obvious for all data analyst that if one component of the measured data behaves in accordance with “fat tail statistics” so all process must be assumed as governed by such statistics. As it was proved.}
  • Figure 1 should be better explained: for example, what does each bar stand for? What does each color means? What does “t” mean? I suggest not taking anything for granted. This will help the correct interpretation of the figure.{-> done. Thank You for this comment}
  • What does the author conclude from the experimental analysis showed in figure 1? {->there is no conclusion. In this figure just experimental data are placed to show result of TFC registration} For example, the author said “We assumed that the period of 5 months during which the measurements were made is sufficient for proper estimation of stationary distribution”: from what can they deduce this?{-> thank You for this comment. Additive explanation was added in the text} Moreover, where is the indication of months in the figure? ?{-> thank You for this comment. Additive explanation was added in the text} And eventually, were the observations deduced from the experimental evaluation based on qualitative considerations? {-> in figure 3}
  • Not all the equations are mentioned in the text. As figures and tables, they should be mentioned and described in the text and then, inserted.{->done though such rules are rare in the publications. The sequence of the formulas is clearly indicated in the text}
  • It seems that many aspects were taken for granted.{-> what aspects for example? There are no such aspects. If You suggest such a thing just show such aspect}
  • I suggest reconsidering the last section: I suggest not calling it summary if it is the discussion section and not inserting a summary section since there is the abstract.{-> Are we talking about the same paper? Which part from summary is in the abstract? I can not agree with this comment at all, or I do not understand this comment}
  • The value of the Hurst exponent for the proposed model cannot be mentioned only at the end of the work.{-> it is first time in figure 3 as a=2H what is quite clear from the text of introduction}.
  • Quality of figures can be improved.{ ->I agree. Done. If paper will be accepted separate files with figures should be added}
  • How does the discussion at lines 207-215 influence the implementation of the proposed model? {-> thank You for this comment. Proper sentences were added}
  • What are the main different results obtained by the Markov process models and by the model based on anomalous diffusion? What is the added value given by the proposed model with respect the previously proposed ones? {-> just carefully read lines 57-76. Common approach with Markov processes is that such processes are gaussian. Basseler and his coauthors proved, that Markov model can be also non-gaussian so with Hurst exponent not equal to one-half. In proposed paper we cite our earlier work with Markov model of the same experiment (we use in here) with really good fit with clinical observations. Here we proved that observed processes have “fat tail” so Bassaler is right. Behavior of medical parameters is in accordance with “fat tail” processes and there is no conflict with Markov model of such behavior. Bassaler work was only theoretical and here we propose an example of real process which we investigate theoretically and experimentally as well.}
  • All the acronyms have to be defined at their first use (the acronym IT, line 56, is not defined, as well as the acronym “FPE” at line 138). {-> thank You for this comment. Proper change was done in line 132 where Fokker-Planck Equation (FPE) is named first time in the text. You are right that IT is slang from information and computer science and is short synonym of “information and computer science”. I have tried to improve it in the text }

  • What does it mean “pdf” at line 172?{-> thank You for this comment. In line 171 I should write “probability density function (pdf)…” , as now it has been done}

Reviewer 2 Report

What was the rationale of consideration of 5 months to be sufficient for stationary distribution

Can TFC be considered as an independent variable, or other Cardia Indexes need to be considered more identifying a more robust evolution model

Effect of history on the observation performed during therapy, if any, appears to be negligibly small:  Can low Hurst exponent completely justify the rationale: Please explain

The authors interpretation of the "evolution of disease" term should be stated clearly in the introduction section (instead of the summary section)

Fig.2 & 3, how were the value of the parameter (alpha) chosen 

Author Response

Comments and Suggestions for Authors with author answers in brace brackets 

What was the rationale of consideration of 5 months to be sufficient for stationary distribution {-> averaged number of hits during 5 month per patient turned out to be almost whole number over the 12 months of observations. So assumption higher than five moths do not carry new results}

Can TFC be considered as an independent variable, or other Cardia Indexes need to be considered more identifying a more robust evolution model {->exactly as it was written – “.. illustrative examples od data..”. It is quite obvious that if one component of the measured independent data behaves in accordance with “fat tail statistics” so all process must be assumed as governed by such statistics. TFC is the reversed change of impedance of chest.}

Effect of history on the observation performed during therapy, if any, appears to be negligibly small:  Can low Hurst exponent completely justify the rationale: Please explain {-> just carefully read lines 57-76. Common approach with Markov processes is that such processes are gaussian. Basseler and his coauthors proved, that Markov model can be also non-gaussian so with Hurst exponent not equal to one-half. In proposed paper we cite our earlier work with Markov model of the same experiment (we use in here) with really good fit with clinical observations. Here we proved that observed processes have  “fat tail” so Bassaler is right. Behavior of medical parameters is in accordance with “fat tail” processes and there is no conflict with Markov model of such behavior. Bassaler work was only theoretical and here we propose an example of real process which we investigate theoretically and experimentally as well}

The authors interpretation of the "evolution of disease" term should be stated clearly in the introduction section (instead of the summary section) {-> thank You for this comment. Done in line 23}

Fig.2 & 3, how were the value of the parameter (alpha) chosen {->thank You for this comment. Wider explanation was added below figure 3}

Submission Date

13 September 2022

Date of this review

24 Oct 2022 22:36:39

Round 2

Reviewer 1 Report

The reviewer would have preferred a division of the sections according to this scheme: "1. Introduction" (in which the author contextualizes the question addressed, presents the state of the art and any disadvantages related to it, and finally the aim of the work); "2. Materials" (with "database" the reviewer meant the description of the data, or collection of data, used, on which the proposed procedure was applied, so the reviewer would suggest not to simply insert a couple of examples of features considered if the quantification of the same represented the amount data actually used to carry out the model); "3. Methods"; "4. Results"; "5. Discussion". The reviewer believes that this kind of discrimination between the various sections could help in better understanding the work, but if this kind of organization is not contemplated by the journal, it remains a suggestion.

The reviewer suggests expressing in the methodological section, the parameters used to quantitatively evaluate the goodness of the proposed model. For example, the RMSE is directly inserted in figure 3 and it does not seem it was mentioned before as parameter used to assess the goodness of the model. This is an example of the fact that some details of the procedure are not clearly expressed.

Referring to lines 83-88 (previously, 77-82), if these lines report what the author has obtained in the work, I encourage to reconsider them, and I suggest not to anticipate in the introduction the results and the conclusions of the study.

Referring to the reviewer’s observation about the summary: summary and abstract sometimes are used as synonyms, so if the author decided to use this term to refer to the last section (title), it could seem that the two sections have the same content. Practically, I suggest using a title for the last section that could not be misunderstood.

I suggest not taking for granted the definition of acronyms. For example, the acronym of root mean square error could be considered quite standard (as acronym of heart rate); nevertheless, it should be defined (as it was defined the acronym of heart rate). 

Author Response

Answers to reviewer comments (in green)

The reviewer would have preferred a division of the sections according to this scheme: "1. Introduction" (in which the author contextualizes the question addressed, presents the state of the art and any disadvantages related to it, and finally the aim of the work); "2. Materials" (with "database" the reviewer meant the description of the data, or collection of data, used, on which the proposed procedure was applied, so the reviewer would suggest not to simply insert a couple of examples of features considered if the quantification of the same represented the amount data actually used to carry out the model); "3. Methods"; "4. Results"; "5. Discussion". The reviewer believes that this kind of discrimination between the various sections could help in better understanding the work, but if this kind of organization is not contemplated by the journal, it remains a suggestion.

{-> I must agree that Your proposal seems create more clear form of text and mine was rather “crosscut”. Some modifications in text have been done. Results are shown separately – thank You for this suggestion once again – as well as title of the last part is “discussion.”}

The reviewer suggests expressing in the methodological section, the parameters used to quantitatively evaluate the goodness of the proposed model. For example, the RMSE is directly inserted in figure 3 and it does not seem it was mentioned before as parameter used to assess the goodness of the model. This is an example of the fact that some details of the procedure are not clearly expressed. {-> thank You for this comment. Some correction in the text was done}

Referring to lines 83-88 (previously, 77-82), if these lines report what the author has obtained in the work, I encourage to reconsider them, and I suggest not to anticipate in the introduction the results and the conclusions of the study. {-> Thank You for this comment. Small correction in abstract as well as in introduction was done.}

Referring to the reviewer’s observation about the summary: summary and abstract sometimes are used as synonyms, so if the author decided to use this term to refer to the last section (title), it could seem that the two sections have the same content. Practically, I suggest using a title for the last section that could not be misunderstood.{-> done. Thank You for this comment}

I suggest not taking for granted the definition of acronyms. For example, the acronym of root mean square error could be considered quite standard (as acronym of heart rate); nevertheless, it should be defined (as it was defined the acronym of heart rate). {->done. Thank You for this comment}

Round 3

Reviewer 1 Report

The author has attempted to “adjust” the structure of the article following the reviewer's suggestions (although the reviewer's suggestions and examples could have been generalized more to all the paper).  The abstract needs to be revised: it should contain the most important aspects of the whole paper, including the results and conclusions (for example, the last sentence of the abstract simply introduces what is present below).

Author Response

The author has attempted to “adjust” the structure of the article following the reviewer's suggestions (although the reviewer's suggestions and examples could have been generalized more to all the paper).  The abstract needs to be revised: it should contain the most important aspects of the whole paper, including the results and conclusions (for example, the last sentence of the abstract simply introduces what is present below). {ANSWER-> kind of “adjust” is the best way to take comments of referee into account in my opinion. I appreciate all comments. Abstract has been changed to more straight description what was done in the paper and why it was done. In my opinion it is enough to get short, condensed resume of the paper. Conclusions are the final part and are placed in the last part of the paper.}
